# Diabetes and Osteoarthritis: Exploring the Interactions and Therapeutic Implications of Insulin, Metformin, and GLP-1-Based Interventions

**DOI:** 10.3390/biomedicines12081630

**Published:** 2024-07-23

**Authors:** Iryna Halabitska, Liliia Babinets, Valentyn Oksenych, Oleksandr Kamyshnyi

**Affiliations:** 1Department of Therapy and Family Medicine, I. Horbachevsky Ternopil National Medical University, Voli Square, 1, 46001 Ternopil, Ukraine; 2Broegelmann Research Laboratory, Department of Clinical Science, University of Bergen, 5020 Bergen, Norway; 3Department of Microbiology, Virology, and Immunology, I. Horbachevsky Ternopil National Medical University, 46001 Ternopil, Ukraine; alexkamyshnyi@gmail.com

**Keywords:** cartilage, insulin resistance, obesity, inflammation

## Abstract

Diabetes mellitus (DM) and osteoarthritis (OA) are prevalent chronic conditions with shared pathophysiological links, including inflammation and metabolic dysregulation. This study investigates the potential impact of insulin, metformin, and GLP-1-based therapies on OA progression. Methods involved a literature review of clinical trials and mechanistic studies exploring the effects of these medications on OA outcomes. Results indicate that insulin, beyond its role in glycemic control, may modulate inflammatory pathways relevant to OA, potentially influencing joint health. Metformin, recognized for its anti-inflammatory properties via AMPK activation, shows promise in mitigating OA progression by preserving cartilage integrity and reducing inflammatory markers. GLP-1-based therapies, known for enhancing insulin secretion and improving metabolic profiles in DM, also exhibit anti-inflammatory effects that may benefit OA by suppressing cytokine-mediated joint inflammation and supporting cartilage repair mechanisms. Conclusions suggest that these medications, while primarily indicated for diabetes management, hold therapeutic potential in OA by targeting common underlying mechanisms. Further clinical trials are warranted to validate these findings and explore optimal therapeutic strategies for managing both DM and OA comorbidities effectively.

## 1. Introduction

Diabetes mellitus (DM) encompasses a variety of disorders that all involve elevated blood glucose levels. The current classification of DM is outlined, with a comparison of the key characteristics of type 1 and type 2 diabetes. Additionally, the criteria for accurate biochemical diagnosis during fasting and oral glucose tolerance tests, as well as the use of hemoglobin A1c (HbA1c), are summarized. The rising prevalence of DM necessitates targeted screening to identify diabetes and prediabetes in at-risk groups. This screening is essential for the early implementation of measures to prevent the onset of diabetes and to slow its progression in these groups [1]. The incidence of DM is rising quickly, often leading to severe metabolic disorders and complications [2]. In recent decades, the prevalence of DM has increased significantly in almost every country, and it can be regarded as a growing epidemic. Urbanization and income status are key factors affecting current prevalence rates, revealing notable differences among various population groups [3].

Type 1 diabetes mellitus (T1DM) is a significant subtype of diabetes, typically diagnosed in youth and characterized by insulin deficiency. The life expectancy of individuals with T1DM has markedly improved over the past three decades due to the availability of exogenous insulin; however, it remains lower than that of the general healthy population [4]. Type 2 diabetes mellitus (T2DM), a prevalent metabolic disorder, arises from two primary factors: impaired insulin secretion by pancreatic β-cells and the reduced responsiveness of insulin-sensitive tissues to insulin [5].

The prevalence of obesity and DM has been steadily increasing worldwide. Both conditions share significant genetic and environmental factors in their development. Obesity enhances the impact of genetic predisposition and environmental influences on DM. The abnormal growth of adipose tissue and the excessive accumulation of certain nutrients and metabolites disrupt metabolic balance through insulin resistance, impaired autophagy, and disturbances in the microbiome–gut–brain axis. This disruption leads to low-grade systemic inflammation, which further destabilizes immunometabolism, accelerates the loss of functional β-cells, and gradually increases blood glucose levels. Due to these complex connections, most treatments for obesity and DM affect both conditions [6].

Osteoarthritis (OA) is recognized as a degenerative joint disease marked by inflammation, chronic pain, and functional impairment [7]. OA is a progressive disease characterized by cartilage degradation, subchondral bone remodeling, and synovial inflammation. The disease is linked to factors such as obesity, mechanical load, and aging. Additionally, various pro-inflammatory immune mediators influence the expression of metalloproteinases, which play a role in cartilage breakdown. Genetic factors also contribute to the susceptibility to OA [8]. The prevailing understanding of osteoarthritis depicts it as a “comprehensive joint ailment”, emphasizing the engagement of not just the articular cartilage but also the synovium, subchondral bone, ligaments, and muscles. Obesity and metabolic syndrome are linked to elevated levels of pro-inflammatory cytokines, heightened secretion of adipokines possessing both protective and detrimental impacts on articular cartilage, an increase in proteolytic enzymes like matrix metalloproteinases and aggrecanases, and a rise in free fatty acids and reactive oxygen species prompted by dyslipidemia [9].

DM and OA are prevalent conditions expected to become even more common [10]. The coexistence of OA and DM is often coincidental, attributed to their high prevalence and shared risk factors [11]. For instance, there is a well-established link between OA and obesity, and most individuals with type 2 diabetes mellitus (T2DM) are also affected by obesity [11,12].

However, there is a lack of information regarding the influence of various forms of diabetes, along with the medications utilized for diabetes treatment, on the progression of osteoarthritis. This review aims to address this gap to offer potential new treatment options and a more comprehensive understanding of the underlying mechanisms. By exploring these connections, this review could both clarify why these diseases often coexist, and also aid in developing future treatments for patients.

## 2. Understanding the Coexistence of Diabetes Mellitus and Osteoarthritis: Pathogenic Links and Therapeutic Consideration

Hyperglycemia is regarded as the primary instigator of joint deterioration by enhancing the production of advanced glycation end products (AGEs), which stimulate chondrocytes and synoviocytes to generate pro-degradative and pro-inflammatory agents, inciting a mild systemic inflammation that triggers local joint inflammation, exacerbating OA progression in different joint components, and leading to neuromuscular impairments that destabilize the joint and exacerbate OA symptoms [13].

In conditions of elevated extracellular glucose levels, the capacity to regulate glucose uptake through the downregulation of glucose transporters is compromised in chondrocytes affected by OA. This leads to the buildup of glucose and increased production of reactive oxygen species (ROS), fostering degenerative alterations and advancing the development of OA [13,14,15].

### 2.1. Intersection of Type 1 Diabetes Mellitus and Osteoarthritis: Shared Mechanisms and Therapeutic Challenges

Type 1 diabetes mellitus (T1DM) impacts 9.5% of the population and is marked by a severe insulin deficiency, resulting in hyperglycemia and various systemic effects. T1DM is considered a potential risk factor for damage and loss of articular cartilage, which could accelerate the onset of OA. The relationship between T1DM and OA remains largely unexplored [16,17].

Furthermore, recent research investigating the relationship between T1DM and OA has yielded conflicting findings, with some studies indicating a positive correlation while others did not. One study conducted histological assessments of joints in T1DM and control subjects, revealing that T1DM mice exhibited measurements of cartilage degeneration consistent with mild OA characteristics. RNA sequencing analyses identified a notable upregulation of genes associated with matrix-degrading enzymes in T1DM, which are known to contribute to cartilage matrix degradation, suggesting their involvement in OA development. Subsequently, the study examined whether preexisting T1DM affects the development of post-traumatic OA following injury. Results at the 6-week mark post-injury revealed that T1DM-injured joints exhibited considerably less cartilage damage and joint degeneration compared to injured non-diabetic joints, indicating a significant delay in the progression of post-traumatic OA. At a cellular level, an increased number of cells expressing chondrocyte markers Col2a1, Acan, and Cytl1 were identified in the T1DM-injured group [17,18].

The significance of glucose metabolism and its derivatives, such as AGEs, sorbitol, and diacylglycerol (DAG), in the pathogenesis of OA and DM is emphasized, as these derivatives activate inflammatory pathways. The potential link between DM and OA is indicated by the inflammatory response due to increased pro-inflammatory cytokine expression [19]. Recent research has illuminated immune cell populations’ temporal dynamics and activation statuses, including macrophages, localized within joints or originating systemically, contributing to inflammatory responses in osteoarthritis. Their complex interactions may explain varying pain and symptom manifestations observed during osteoarthritis exacerbations. Additionally, investigations into biological and environmental factors such as exercise, age, and diet have explored their potential roles in mitigating or exacerbating osteoarthritis-related inflammation. However, despite these advancements, effective disease-modifying treatments targeting inflammation in osteoarthritis have yet to be developed [20]. Mitochondrial dysfunction, characterized by impaired mitophagy resulting in the release of mitochondrial reactive oxygen species (mtROS) and mitochondrial DNA (mtDNA), plays a critical role in initiating inflammation in T1DM. This process involves upregulating pro-inflammatory cytokines and engaging receptors akin to those involved in pathogen-associated responses. Furthermore, mtROS and mtDNA activate pathways that contribute to the progression of chronic inflammation, which is closely linked to autoimmunity in T1DM [21]. Therapeutic agents capable of influencing inflammation show promise for both T1DM and OA.

### 2.2. Insulin Resistance, Obesity, and Osteoarthritis: Intersecting Pathways and Clinical Implications

Obesity is linked to various diseases, particularly insulin resistance and T2DM. Evidence-based studies indicate that adipose tissue (AT) is highly adaptable in its metabolic functions, responding to the body’s energy needs and managing the balance between fasting and feeding throughout the day. It also adjusts to long-term changes in energy balance through tissue expansion and reduction [22,23,24]. This adaptability, especially the ability to expand and contract, is crucial for AT health and overall metabolic balance, and changes in these responses may contribute to the varying metabolic health seen in people with obesity [25,26,27]. A significant discovery in mice revealed that AT produces pro-inflammatory cytokines, which lead to insulin resistance, and that AT macrophages accumulate in obese individuals, supporting the hypothesis that adipose inflammation is a key driver of insulin resistance in obesity [28,29,30]. Although there is a marked increase in inflammatory macrophages and pro-inflammatory protein gene expression in the subcutaneous abdominal AT of individuals with metabolically unhealthy obesity compared to those with metabolically healthy obesity, it remains challenging to determine if this inflammation is a cause or effect of insulin resistance [31,32,33]. Metabolically healthy obesity denotes a state in which individuals possess excessive body fat without manifesting the usual metabolic dysfunctions linked to obesity, such as insulin resistance, dyslipidemia, or hypertension [34]. Metabolically unhealthy obesity is characterized by the presence of metabolic dysfunctions such as insulin resistance, dyslipidemia, or hypertension in individuals with excess body fat, indicating heightened health risks associated with obesity [35].

The concentration of free fatty acids associated with obesity and T2DM can negatively impact pancreatic beta cells. Basal levels of plasma free fatty acids contributed to hyperinsulinemia in normoglycemic obese patients [36,37,38]. There is a strong link between obesity and increased rates of free fatty acids in the bloodstream, which are then delivered to body tissues [39,40,41]. Although numerous studies show that elevated plasma free fatty acid levels are a significant cause of liver and muscle insulin resistance, conflicting data from real-world scenarios challenge these findings. Several studies indicate that the breakdown of AT triglycerides is highly sensitive to insulin [42,43]. Postprandial suppression of lipolysis and plasma free fatty acid concentrations is generally similar in both lean and obese individuals, as the greater postprandial increase in plasma insulin in obese individuals may compensate for their increased fat mass [44,45,46]. The relationship between insulin resistance and obesity remains complex and requires a clear understanding of the pathways linking T2DM to the increase in inflammatory macrophages in subcutaneous adipose tissue.

Obesity is the most significant risk factor for the onset and progression of osteoarthritis, with recent research highlighting additional contributing factors such as adipose tissue accumulation, insulin resistance, and the misalignment of innate and adaptive immune responses, wherein various inflammatory cells, particularly polarized macrophages and their mediators, play a crucial role in the pathological changes of the synovial joint [37]. Obesity, a major and modifiable risk factor for osteoarthritis, not only increases mechanical stress on tibiofemoral cartilage but also correlates with higher OA prevalence in non-weight-bearing areas due to its role in systemic inflammation, driven by adipose tissue-derived cytokines and adipokines like adiponectin and leptin, which regulate inflammatory immune responses and contribute to elevated levels of pro-inflammatory cytokines such as TNF-α, IL-1β, and IL-6, produced by adipose tissue macrophages [47]. In individuals suffering from knee osteoarthritis and exhibiting overweight or obesity, dietary adjustments and exercise, when compared to an attention control group, resulted in a statistically significant albeit modest reduction in knee pain over an 18-month period [48]. Nutritional interventions can potentially impact adipose tissue mass and the secretion of inflammatory mediators, which may, in turn, exert effects on other tissues in the body, including bone and articular cartilage [49]. Emphasizing BMI in osteoarthritis research could potentially perpetuate weight bias in clinical settings and exacerbate disparities in accessing effective treatments for osteoarthritis [50].

### 2.3. Type 2 Diabetes Mellitus and Osteoarthritis: Synergistic Impact on Musculoskeletal Health and Treatment Strategies

Numerous studies have documented the increased occurrence of OA in patients with diabetes. Meta-analyses have confirmed an epidemiological link between T2DM and OA, indicating that individuals with diabetes have a higher risk of developing OA [51,52,53]. However, the strength of this association can differ based on factors such as age, ethnicity, duration of T2DM, body weight, and the specific joints affected by OA [54].

Various studies have shown a link between long-term T2DM and faster progression of OA, with increased rates of synovial inflammation and joint pain [55,56]. This connection is even stronger in younger diabetic individuals with hand OA, who are more likely to develop the erosive form of the disease [57,58]. Interestingly, the relationship between T2DM and OA appears to be bidirectional. A cohort study found that joint pain and reduced mobility in the knee and hip, leading to a sedentary lifestyle, significantly increased the risk of developing T2DM in individuals over 55 years of age [59,60].

Traditionally, age-related joint degeneration and biomechanical stress from being overweight were seen as the primary risk factors for OA in diabetic individuals. However, recent advancements in understanding OA and T2DM have highlighted the influence of systemic factors such as dyslipidemia, hyperglycemia, and inflammation—collectively known as metabolic syndrome—that may directly contribute to OA. This has led to the recognition of a new clinical phenotype called metabolic OA [61,62,63]. This form of OA affects both load-bearing joints (like the hip and knee) and non-load-bearing joints (such as the hand), indicating that factors beyond just biomechanical stress are at play [64,65]. T2DM and OA are interconnected through the chronic systemic inflammation associated with metabolic syndrome. Under hyperglycemic conditions, OA patients’ chondrocytes fail to downregulate glucose transport [66,67,68]. High glucose levels trigger the production of ROS in OA cartilage [69]. The catabolic activity of ROS generates inflammatory mediators like IL-1β and NF-κB, which lead to chondrocyte degradation and apoptosis, thus damaging the chondrocytes [70,71]. Additionally, OA chondrocytes in a hyperglycemic environment express higher levels of matrix metalloproteinases than normal chondrocytes [72,73,74]. An in vivo cohort study found that elevated fasting serum glucose levels are linked to increased cartilage damage, indicated by bone marrow lesions and loss of tibial cartilage volume, particularly in post-menopausal women compared to men [75]. This gender disparity may stem from estrogen levels, which are known to have a protective effect on cartilage. These findings highlight the detrimental effects of hyperglycemia on articular cartilage and suggest that disrupted glucose metabolism may directly link OA and T2DM [76]. Another harmful effect of hyperglycemia is the induction of AGEs [77,78]. The age-related accumulation of AGEs in articular cartilage creates a pathogenic environment, leading to symptoms of OA, such as stiffness and cartilage degradation. High glucose levels in diabetics result in increased AGEs formation. AGEs and their receptor initiate the inflammatory cascade, primarily through the production of pro-inflammatory TNF-α and the activation of the transcription factor NF-κB [79,80].

Human chondrocytes have functional insulin receptors that respond to physiological insulin levels, but the expression and activity of these receptors are lower in OA chondrocytes compared to normal chondrocytes [81,82,83]. Insulin treatment increases the expression of metalloproteinases-13 and IL-1β, and reduces autophagy, a crucial homeostatic process, in chondrocytes by decreasing LC3 II expression and increasing phosphorylation of Akt and rpS6. This suggests that the excess insulin seen in T2DM patients may harm cartilage and contribute to OA [11]. Insulin is an essential negative regulator of synovial inflammation and catabolism, so the development of insulin resistance in obese individuals would impair insulin’s ability to suppress the production of inflammatory and catabolic mediators that promote OA [84,85].

## 3. The Potential Impact of Diabetes Therapies on Osteoarthritis

The management of diabetes mellitus, particularly T2DM, often necessitates a multifaceted pharmacological approach to achieve optimal glycemic control and mitigate complications. Insulin therapy, fundamental for both T1DM and advanced T2DM, compensates for insufficient endogenous insulin production, helping to regulate blood glucose levels with various formulations tailored to address basal and prandial needs. Metformin, a first-line oral anti-hyperglycemic agent, enhances peripheral glucose uptake, decreases hepatic glucose production through AMPK activation, and improves insulin sensitivity while also reducing cardiovascular events and mortality. GLP-1 agonists mimic the incretin hormone GLP-1 to enhance glucose-dependent insulin secretion, suppress glucagon release, slow gastric emptying, and promote satiety, contributing to both glycemic control and weight loss [86].

The primary objectives in managing OA include mitigating pain, improving joint mobility, and maintaining overall joint function. Recent strides in comprehending OA’s underlying pathophysiology have spurred investigations into a wide array of therapeutic strategies, including advancements in tissue engineering, manipulation of the immune system, refinement of surgical techniques, and the development of pharmacological and non-pharmacological treatments. However, despite these advancements, a definitive cure for OA remains elusive, underscoring the need for personalized treatment approaches tailored to the specific stage and manifestations of the disease [87]. Focusing on BMI in osteoarthritis research has the potential to perpetuate weight bias within clinical practice settings, influencing treatment decisions and patient outcomes based on weight alone. This approach may exacerbate disparities in accessing effective treatments for osteoarthritis, particularly for individuals with higher BMIs who may face barriers to receiving optimal care. Addressing these biases is crucial to ensure equitable healthcare delivery and improved outcomes for all patients with osteoarthritis [50].

### 3.1. Insulin Use and Osteoarthritis: Evaluating Effects and Therapeutic Implications

It has been observed that insulin, either independently or in conjunction with inflammatory factors, can promote synovial inflammation during the advancement of osteoarthritis. Insulin demonstrates significant capability in enhancing the inflammatory characteristics of fibroblast-like synoviocytes (FLSs), increasing cell viability, and boosting the production of inflammatory cytokines. Additionally, insulin fosters chemokine production and augments macrophage chemotaxis. Moreover, insulin activates the PI3K/mTOR/Akt/NF-ĸB signaling pathway while concurrently inhibiting autophagy in FLSs. Data suggest that preblocking three specific signaling pathways with pathway inhibitors in FLSs significantly diminishes insulin-induced inflammatory responses. Furthermore, insulin is shown to elevate levels of inflammatory cytokine receptors in FLSs, with PI3K/mTOR/Akt/NF-ĸB signaling inhibitors capable of reversing this effect. Notably, insulin sensitizes synovial inflammation mediated by inflammatory factors (including metalloproteinases production and activation of intracellular signaling pathways). Collectively, these findings suggest that insulin may exacerbate synovial inflammatory conditions, thereby contributing to the progression of OA [88,89]. Insulin might stimulate the generation of several pro-inflammatory substances (such as interleukins, tumor necrosis factor-alpha, and metalloproteinase-13) linked with OA [11]. In vitro studies have shown that insulin has the potential to hinder chondrocyte maturation and enhance cartilage degradation, thereby exacerbating the pathological progression of OA [90]. It was indicated that the PI3K/AKT and mTOR signaling pathways play a role in the pathophysiological impacts of insulin in OA [81,90]. NF-ĸB is responsible for regulating the expression of inflammatory factors and metalloproteinases within the joints, and it is widely recognized to have a crucial involvement in the development of OA (Figure 1) [88,91,92].

The research has demonstrated that insulin can induce the loss of proteoglycan components, and elevate levels of inflammatory cytokines and metalloproteinases in chondrocytes, and these effects may be attributed to the inhibition of chondrocyte autophagy. Furthermore, clinical observations from this study revealed decreased autophagy in the knee cartilage of diabetic patients compared to non-diabetic counterparts, evidenced by a significant reduction in the expression of the autophagy-related protein LC3II. Notably, the study unveiled that rapamycin, acting as an autophagy activator and an inhibitor of the mTOR signaling pathway, can mitigate insulin-induced suppression of autophagy and subsequent cartilage degradation [90]. The research has indicated that the biological effects of insulin vary depending on the dose and the specific type of cell involved [93]. In addition to the effects on FLSs observed in this study, insulin was also found to impact chondrocytes [90,93]. However, insulin’s role in chondrocyte differentiation has been reported to be either promotive or inhibitory [93,94], thereby either improving or worsening cartilage degeneration depending on its concentration [90]. Moreover, at the molecular level, insulin’s influence varies significantly when its concentration is extremely high or low [95,96,97]. It was discovered that low and supraphysiological insulin levels have varying impacts on aggrecan and proteoglycan synthesis in chondrocytes, likely due to different insulin receptors. Additionally, it was found that high insulin levels (1 μM) decreased FoxO transcriptional activity, while low insulin levels (0.1 μM) also reduced FoxO transcriptional activity in SZ95 sebocytes in vitro [97]. It was demonstrated that insulin selectively influences specific downstream responses of the Akt pathway in a dose-dependent manner. Therefore, it was concluded that varying insulin concentrations are linked to different mechanisms of insulin action, which can modulate cellular responses [95,98]. Previous studies have indicated that specific cellular responses can be triggered only by high concentrations of insulin [99,100,101]. Collectively, these studies highlight that different effector cells involved in the complex pathophysiological processes of various diseases exhibit distinct sensitivities to insulin [99,100,101].

Immune cells require glucose for energy production [102]. Like adipose, muscle, and liver cells, they have insulin receptors (IRs) on their surfaces [103,104]. Insulin, functioning as a glucose-regulating hormone through IRs, also acts as a growth factor and cytokine regulator, thereby influencing immune modulation [105,106,107]. Insulin influences the immune response both indirectly by lowering glucose levels and directly by affecting immune cells, impacting their proliferation and signal transduction [108,109]. High blood sugar negatively impacts the immune system by causing cell stress and producing AGEs and ROS, which trigger the release of pro-inflammatory mediators. Thus, insulin’s role in lowering glucose can reduce “glucose toxicity” and cell stress, providing an anti-inflammatory effect [110,111].

Insulin, beyond its metabolic role, exerts anti-inflammatory effects via PI3K/Akt pathway activation, suppressing TLR4 signaling and NF-κB activity in leukocytes, thus modulating immune responses and inflammation [112,113,114].

The precise regulation of insulin secretion by pancreatic β-cells is crucial for maintaining metabolic balance. β-cell mass must dynamically adjust to metabolic demands and can undergo significant changes in response to various conditions. The mTOR complexes, specifically mTORC1 and mTORC2, play pivotal roles in modulating β-cell mass. In states of systemic insulin resistance, mTORC1/mTORC2 signaling in β-cells is essential for increasing β-cell mass and enhancing insulin secretion. However, the failure of these compensatory mechanisms contributes to the development of type 2 diabetes, highlighting the complex and still incompletely understood role of mTOR complexes in β-cell dysfunction [115,116].

Some studies indicate insulin’s potential pro-inflammatory role, affecting PMN leukocyte functions without increasing ROS production [117,118]. Insulin reduces ROS production in monocytes and dose-dependently inhibits tissue factor procoagulant activity via regulatory mechanisms [119,120].

While IRs are found on the surface of B cells, monocytes, and resting neutrophils, they are not present on resting T cells [121,122]. However, IR expression is significantly increased on activated T cells [123,124], which is crucial for meeting the high glucose demand necessary for T cells to achieve full effector functions. Insulin signaling in T cells enhances their activation by promoting protein synthesis, glucose uptake, and amino acid transport [123]. It was demonstrated in vitro that insulin shifts the response toward Th2, reducing the Th1 to Th2 ratio. This shift results in a change in cytokine secretion, with a decreased interferon-gamma to IL-4 ratio and increased phosphorylation of extracellular signal-regulated kinase (ERK), one of the four MAPK signaling pathways [125,126]. Experiments on mice lacking IRs demonstrated impaired polyclonal activation of CD4+ T cells, as well as deficiencies in cytokine production, migration, and proliferation [127]. Similar impairments were observed in CD8+ T cells, which showed reduced cytotoxicity in response to alloantigens. Studies on obese patients have shown that insulin resistance and related disorders are characterized by a cytokine imbalance, with elevated levels of TNF-α, IL-6, IL-1β, CRP, and NF-κB [128]. Th17 and Treg cells are two subsets of CD4+ T cells that share some developmental pathways but have different phenotypes and opposite functions. Th17 cells are pro-inflammatory, while Treg cells are anti-inflammatory [129]. An altered balance between Treg and Th17 cells is implicated in arthritis and other immune-mediated conditions [130]. Activation of quiescent T cells occurs through the stimulation of the T cell receptor (TCR) complex and the binding of the co-receptor CD28 to co-stimulatory molecules (Acuto). TCR engagement triggers intracellular signaling via the ERK/MAPK pathways, while CD28 signaling activates the PI3K-Akt-mTOR pathway [131,132].

PI3K-Akt signaling promotes glycolysis and increases the expression of glucose transporter 1 (Glut1), thereby enhancing glucose uptake. Overexpression of Glut1 facilitates the differentiation of T follicular helper (Tfh) cells, a T cell subset involved in B cell regulation, which may contribute to autoimmunity in both type 1 diabetes and arthritis [133,134]. Additionally, PI3K-Akt activation leads to mTOR activation, which supports the differentiation of Th1, Th17, and Tfh cells [135]. Moreover, mTOR can inhibit the formation of long-lived Tregs while favoring effector Tregs [136]. Tregs lacking mTOR exhibit reduced frequency, resulting in spontaneous activation of effector T cells and inflammation [137]. AMPK can inhibit cellular growth by suppressing the mTORC1 pathway [138]. Activation of AMPK and disruption of mTOR signaling have been shown to reduce inflammation in experimental arthritis. AMPK’s control over fatty acid metabolism can also influence cell fate decisions in CD4+ T cells, particularly affecting the balance between Th17 and Treg lineages [139,140]. Additionally, growth factors like insulin, IGF-1, and IL-2 can stimulate PI3K-Akt-mTOR signaling. Insulin and insulin-like growth factors (IGFs) utilize common PI3K-AKT-mTOR and RAS-RAF-MEK-ERK pathways. Activation of IGF-1 receptor (IGF1R) promotes Akt-mTOR signaling, enhances glycolysis, and favors Th17 differentiation, impacting inflammatory processes such as arthritis through IL-6 modulation [141].

In insulin resistance, Akt signaling becomes impaired, leading to the hyperactivation of mTORC1 and increased glycolysis. This heightened glycolysis in macrophages impacts their responses to pathogens and danger signals [142]. Insulin significantly boosts the LPS-dependent expression of IL-1β and IL-8, as well as the induction of enzymes involved in prostaglandin E2 (PGE2) synthesis by macrophages [143]. Both in vivo and in vitro studies suggest that insulin restores phagocytosis and promotes phagocytosis-induced apoptosis in neutrophils. Additionally, insulin treatment prompts macrophages to transition from an M1 to an M2 polarization state [142].

Previous research has highlighted the significance of insulin signaling in the biology and pathology of the joint, particularly in its ability to regulate bone architecture by affecting osteoblasts and osteoclasts [144,145,146]. In vitro experiments have shown that insulin increases IR expression and stimulates cell proliferation and differentiation in MG-63 cells through the MAPK and PI3K pathways, leading to enhanced alkaline phosphatase activity, secretion of type I collagen, and expression of osteocalcin [147,148]. Activation of mTORC1 by insulin-like growth factor 1 (IGF-1), released during bone resorption, promotes osteoblast differentiation of mouse bone marrow stromal cells (BMSCs), playing a critical role in the transition from pre-osteoblasts to mature osteoblasts [149,150].

However, insulin also affects osteoclasts. Through the ERK1/2 pathway, insulin upregulates receptor activator of nuclear factor-kB (RANK), contributing to enhanced osteoclast differentiation by RANK ligand (RANKL) [151]. The precise effects of mTORC1 on osteoclasts are not fully understood. Deletion of raptor, leading to mTORC1 inactivation in osteoclast precursors, or activation of mTORC1 by deletion of tuberous sclerosis complex 1 (Tsc1), can, respectively, increase or decrease osteoclastogenesis. Mechanistically, this is attributed to mTORC1’s inhibition of NF-kB and nuclear factor of activated T cells 1 (NFATc1), both critical transcription factors for osteoclastogenesis [152]. Furthermore, RANKL-dependent osteoclastogenesis is impaired in Tsc1-deficient bone marrow macrophages, where TSC1 negatively regulates mTORC1 [153]. It was suggested that mTORC1 inhibition by rapamycin treatment or genetic deletion suppressed in vitro osteoclast differentiation, which was rescued by upregulation of the mTOR downstream target S6K1 [154].

Insulin resistance and hyperinsulinemia have been implicated in the development of OA and metabolic syndrome [155,156]. In human chondrocytes, insulin dose-dependently activates the mTOR signaling pathway and phosphorylates Akt, resulting in impaired cellular autophagy, a crucial mechanism for removing and degrading damaged intracellular components [90]. Additionally, insulin decreased the content of proteoglycans and increased the expression of metalloproteinase-13 and IL-1β, both of which play significant roles in chondrocytes and in the degradation of cartilage [90,91].

### 3.2. Metformin’s Influence on Osteoarthritis: Mechanisms and Therapeutic Implications

Since diabetic patients face a higher risk of bone degradation, anti-diabetic medications may offer protective effects against bone disorders [157,158]. Metformin, an oral anti-hyperglycemic drug and the first-line treatment for T2DM, primarily works by inhibiting hepatic gluconeogenesis. Metformin, through the activation of AMPK, inhibits mTOR, which plays a crucial role in regulating lymphocyte immunometabolism and the balance of pro-inflammatory and anti-inflammatory cell populations within joints. When mTOR is inhibited, the production of pro-inflammatory Th1, Th17 cells, and M1 macrophages decreases, leading to a predominance of anti-inflammatory Treg cells and M2 macrophages (Figure 2). Therefore, the inhibition of mTOR via AMPK activation by metformin may have potential therapeutic effects in the treatment of inflammatory diseases, reducing the activity of pro-inflammatory cells while promoting the predominance of anti-inflammatory cell populations [159].

The ability of metformin to regulate immune responses and improve gut microbiota diversity presents an encouraging opportunity for therapeutic interventions in individuals with type 2 diabetes who are at a higher risk of experiencing severe outcomes from COVID-19 [160,161]. The influence of type 2 diabetes, metformin, and insulin on COVID-19 was individually assessed. Among patients who received metformin, the CRP level was notably reduced compared to those who did not receive metformin [162,163,164]. Metformin’s ability to influence immune responses and improve gut microbiota diversity indicates a promising path for therapeutic strategies in individuals with type 2 diabetes [163,165]. The administration of the type 2 diabetes medication metformin holds promise for treating this comorbidity, as it not only lowers blood sugar levels but also boosts the population of gut bacteria that stimulate regulatory T cell responses [166,167].

Metformin targets mitochondria, which produce ATP through oxidative phosphorylation [168]. This process generates ROS, which can cause oxidative stress and mitochondrial dysfunction, both associated with insulin resistance in skeletal muscle, liver, fat, and pancreas [158,169].

Metformin’s metabolic effects are mainly due to its inhibition of the mitochondrial respiratory chain (complex 1), leading to ATP depletion and increased cytosolic AMP production [170]. This indirectly activates AMPK by phosphorylating Thr-172 in its alpha subunit, reducing gluconeogenesis in the liver. Elevated AMP levels also inhibit adenylate cyclase, decreasing cAMP production. Consequently, protein kinase A activity and its target, cyclic AMP response element binding protein, are inhibited, lowering fasting glucose levels [171,172].

Beyond reducing hepatic glucose production, metformin enhances insulin sensitivity by inhibiting lipogenesis, increases peripheral glucose uptake through GLUT4 enhancer factor phosphorylation, and reduces insulin-induced suppression of fatty acid oxidation [173,174,175]. Additionally, metformin mitigates chronic inflammation through its anti-inflammatory properties and promotes autophagy by inhibiting mTOR phosphorylation via AMPK activation [176]. Individuals with T2DM are at a higher likelihood of experiencing hand or knee OA compared to those without diabetes. Conversely, individuals with OA have an increased risk of developing T2DM compared to age- and sex-matched counterparts without OA.

Metformin, along with weight loss, shows promise as a disease-modifying treatment for knee osteoarthritis in obese patients, potentially reducing cartilage loss and the need for knee replacement surgery [177].

Metformin administration, initiated before or after destabilization of the medial meniscus (DMM) surgery, significantly attenuated cartilage degradation as evidenced by decreased Osteoarthritis Research Society International scores and preserved cartilage areas, associated with upregulated AMPK expression in articular cartilage tissue [178].

Various animal models suggest metformin’s potential therapeutic impact on OA, reducing cartilage degradation and modulating pain via AMPK activation [179]. Metformin’s chondroprotective effect involves upregulating AMPKα1 expression, demonstrated in genetically modified and DMM-induced OA mice, suggesting therapeutic potential via AMPK/mTOR pathway modulation [180]. Metformin is shown to activate AMPK and SIRT1 pathways, protecting chondrocyte mitochondrial function and potentially preventing OA development clinically [181]. Metformin attenuated IL-1β and TNF-α induced NO and MMP release [182]. Diabetes mellitus, especially type 2, increases skeletal complications and osteoarthritis risk due to hyperglycemia and advanced glycosylation end products, addressed by metformin’s bone-protective effects through AMPK [183].

Mesenchymal stem cells (MSCs) possess multilineage differentiation potential and mitigate cartilage degradation through immunomodulatory functions. Metformin-enhanced adipose tissue-derived human MSCs show promising chondroprotective and analgesic effects in osteoarthritis, highlighting their therapeutic potential [182,184]. The study investigated metformin’s impact on osteoporotic and normal fracture healing, demonstrating its ability to accelerate healing and promote angiogenesis through HIF-1α upregulation and YAP1/TAZ inhibition, crucial for type H vessel formation [185]. Metformin attenuates IL-1β-induced OA inflammation via SIRT3/PINK1/Parkin signaling, enhancing mitophagy for mitochondrial function [186,187]. The study suggests AMPK and GDF-15 as potential OA therapies, warranting randomized controlled trials for metformin’s efficacy [188]. Metformin’s pharmacological activity relies on organic cation transporters (OCTs) for tissue penetration and therapeutic efficacy [189,190,191]. Metformin use showed genetic protection against HER-positive breast cancer, involving testosterone levels [192]. Metformin’s efficacy and oral bioavailability depend on transporters [193,194]. Various methods have been explored for improving metformin delivery for musculoskeletal therapies [195,196,197].

### 3.3. The Role of GLP-1-Based Therapies in Osteoarthritis: Mechanisms and Potential Benefits

Glucagon-like peptide-1 (GLP-1) and glucose-dependent insulinotropic polypeptide trigger insulin release from pancreatic β cells in response to glucose levels. However, the rapid degradation of native GLP-1 by dipeptidyl peptidase 4 (DPP-4) limits its clinical effectiveness. Consequently, GLP-1 analogues such as liraglutide, exenatide, semaglutide, and lixisenatide, engineered to resist DPP-4 cleavage, are now utilized for managing T2DM [198,199]. GLP-1 agonists are crucial for treating type 2 diabetes and obesity, delaying gastric emptying significantly for glycemic control and weight loss [200]. GLP-1 exerts insulinotropic effects and exhibits anti-inflammatory properties beneficial to the brain, heart, and lungs. GLP-1 receptors are abundant in various tissues, including the pancreas, intestine, and central nervous system [201,202,203]. Patients on long-acting GLP-1 receptor agonists like semaglutide face aspiration risks during anesthesia [204]. GLP-1 analogues show promise for OA due to their anti-inflammatory effects and presence of GLP-1 receptors in joint tissues [205,206]. GLP-1 receptor agonist therapies, through their potential to induce weight loss, may exert disease-modifying effects on knee OA in individuals with comorbid T2DM [207]. GLP-1’s consistent efficacy in reducing food intake and body weight spans across obese individuals, including adolescents and adults. Its mechanism via a single G protein-coupled receptor, coupled with extensive safety data in T2DM patients, supports long-term use for obesity and associated conditions like cardiovascular disease and NASH. Advances suggest GLP-1 therapies may rival bariatric surgery in managing obesity and its complications [208,209].

Drugs that reduce low-grade systemic inflammation might also act locally in the joints [210]. Therefore, incretinomimetics that activate the GLP-1R pathway could be a promising approach for treating OA.

The role of the GLP-1R signaling pathway in chondrocytes has begun to be explored and requires further investigation. Immunohistochemistry detected GLP-1R in normal and OA articular chondrocytes in rat knee sections. GLP-1R signaling is linked to preventing apoptosis, anti-inflammatory activity, and matrix protection [211,212].

Liraglutide protects rat chondrocytes by activating the PI3K/Akt pathway, reducing ER stress-induced apoptosis, increasing Bcl-2, and decreasing cleaved caspase 3 levels. This effect was validated in an ACL rat model [211].

Currently, GLP-1 receptor agonists (GLP-1 RAs) are available in various formulations, including daily injections and a recently approved daily oral preparation of semaglutide, showing efficacy comparable to weekly injections. They share mechanisms such as enhancing insulin secretion, suppressing glucagon release, slowing gastric emptying, reducing post-meal glucose spikes, and promoting weight loss. GLP-1 RAs are recommended as initial injectable therapy for T2DM due to their efficacy in glucose control, weight reduction, and cardiovascular benefits, particularly in high-risk patients with cardiovascular disease. Ongoing research explores their potential in other conditions like type 1 diabetes and neurodegenerative diseases, suggesting a broadening role beyond diabetes management [213].

Activation of GLP-1R inhibits NF-κB, crucial in inflammation and cell regulation [214]. Co-agonist therapies like tirzepatide and amylin combinations show strong clinical promise, enhancing the weight loss potential of GLP-1R agonists like semaglutide [215]. In TNF-activated human chondrocytes and thapsigargin-induced rat chondrocytes, suppressing the NF-κB pathway resulted in decreased release of inflammatory mediators like IL-6, CCL2, and TNF [211]. All GLP-1 RAs improved HbA1c in a 12-week study among Japanese individuals with type 2 diabetes, highlighting varied mechanisms in glucose control and weight loss [216]. In primary mouse chondrocytes, administration of liraglutide decreased the mRNA expression of iNOS, MMP-13, and ADAMTS5, resulting in reduced secretion of inflammatory substances such as nitric oxide, prostaglandin E2, and IL-6 [210]. In the rat model of inflammatory osteoarthritis induced by monoiodoacetate (MIA), activation of GLP-1 receptors initiated the PKA/CREB signaling pathway, leading to a reduction in inflammation within cartilage [217]. Basic scientific studies revealed that GLP-1 analogs exert immunomodulatory effects independent of weight, inhibiting the NF-κB pathway through specific molecular mechanisms in arthritis [218]. GLP-1 analogues demonstrate anti-catabolic effects by decreasing the expression of key enzymes involved in cartilage degradation in response to TNF stimulation. This preservation of extracellular matrix components like aggrecan and type II collagen suggests potential benefits for ОА therapy. Additionally, alterations in the phospholipid layer covering the cartilage surface can disrupt joint function and contribute to ОА pathogenesis [211,219].

Semaglutide use during total knee arthroplasty reduced sepsis and joint infections but increased myocardial infarction, acute kidney injury, pneumonia, and hypoglycemia risks [220].

GLP-1R expression has been identified in human monocyte-derived macrophages and the murine cell line RAW264.7, but research on GLP-1/GLP-1R signaling in macrophages is limited [221,222]. GLP-1 RAs are approved for diabetes and obesity treatment; they also exhibit anti-inflammatory properties across various tissues and pathways [223].

GLP-1R activation modulates macrophage polarization through PKA/CREB signaling, reducing JNK phosphorylation and enhancing STAT3 phosphorylation, influencing immune responses [221,224,225]. This pathway is critical in murine models for promoting M2 macrophage differentiation, enhancing immune modulation and tissue repair processes [226,227,228]. Preclinical and clinical studies demonstrate GLP-1 RAs’ cardioprotective effects, efficacy in hypertension and dyslipidemia, substantial weight loss in diabetes and obesity, and neuroprotective roles in stroke and neurodegenerative diseases. However, manageable adverse effects include gastrointestinal symptoms, increased heart rate, and potential renal issues [229]. In inflamed synovium, GLP-1R activation in macrophages shifts them from the M1 to the M2 phenotype, decreasing IL-6, TNF-α, and iNOS mRNA expression. This suggests GLP-1 therapies could mitigate inflammation by reducing macrophage infiltration and adhesion molecule expression [62,230]. The study investigates lixisenatide’s GLP-1 receptor agonism effects on arthritis pathology in human fibroblast-like synoviocytes, marking the first exploration of this treatment’s impact in this context [231]. Studies illustrate liraglutide’s role in inhibiting lipid accumulation and oxidative stress triggered by oxidized low-density lipoprotein in macrophages, mediated through GLP-1R pathway activation [232,233].

GLP-1based therapy has emerged as a promising treatment for osteoarthritis, targeting both metabolic and inflammatory pathways involved in the disease’s progression. Studies have demonstrated that GLP-1 agonists can reduce inflammation in the synovial membrane and improve cartilage integrity. Additionally, GLP-1 therapy may aid in weight management, thereby alleviating joint stress and further mitigating osteoarthritis symptoms.

## 4. Conclusions

DM and OA are prevalent chronic conditions associated with significant morbidity and healthcare burden worldwide. Treatment agents for DM have the potential to influence the progression of OA. Insulin, as a key regulator of glucose metabolism, exhibits potential dual roles in OA by influencing cartilage homeostasis and inflammatory responses within joint tissues. Metformin, renowned for its glucose-lowering effects via AMPK activation, also shows promise in mitigating OA progression through its anti-inflammatory properties and potential preservation of cartilage integrity. Additionally, GLP-1-based therapies, which enhance insulin secretion and improve glycemic control in DM, may exert protective effects in osteoarthritis by modulating inflammation, promoting cartilage repair mechanisms, and potentially slowing joint degeneration. Further clinical studies are warranted to elucidate the precise mechanisms and therapeutic efficacy of these agents in OA management, paving the way for integrated treatment strategies targeting both DM and OA comorbidities.

## Figures and Tables

**Figure 1 biomedicines-12-01630-f001:**
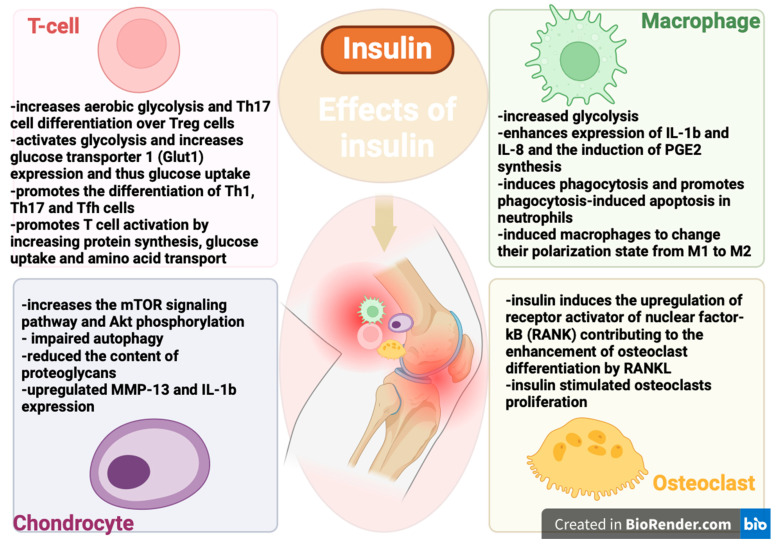
Illustration depicting the multifaceted effects of insulin on immune cells (T cells and macrophages), chondrocytes, and osteoclasts, emphasizing its regulatory role in immune response modulation, cartilage maintenance, and bone metabolism. Th17—T helper 17 cells, Glut1—Glucose transporter 1, mTOR—mammalian target of rapamycin, MMP-13—Matrix metalloproteinase-13, IL-1β—Interleukin-1 beta, IL-8—Interleukin-8, PGE2—Prostaglandin E2, M1—M1 macrophages (classically activated macrophages), M2—M2 macrophages (alternatively activated macrophages), RANK—Receptor activator of nuclear factor kappa-B, RANKL—RANK ligand. Figure 1 has been created in BioRender.com (accessed on 2 July 2024).

**Figure 2 biomedicines-12-01630-f002:**
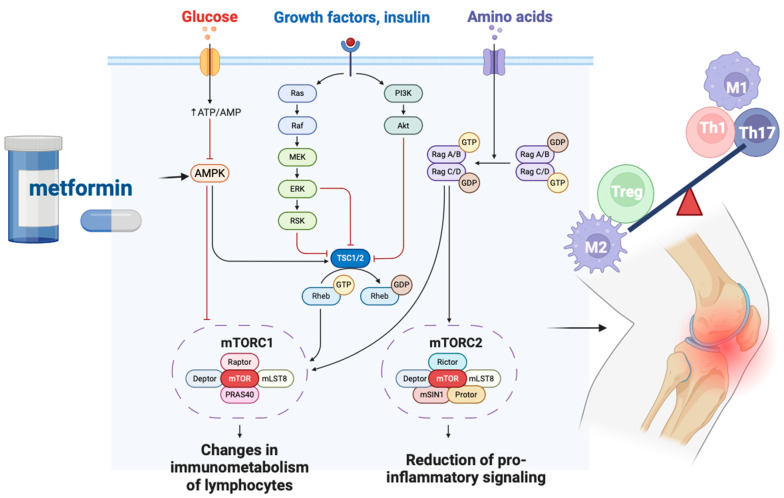
Metformin’s activation of AMP-activated protein kinase (AMPK) leads to the inhibition of mTOR (mammalian target of rapamycin), a pivotal regulator of lymphocyte immunometabolism and the equilibrium between pro-inflammatory and anti-inflammatory cell populations within joint tissues. This inhibition results in decreased production of pro-inflammatory Th1 and Th17 cells, along with M1 macrophages, thereby promoting a predominance of anti-inflammatory Treg cells and M2 macrophages. ATP—Adenosine triphosphate, AMP—Adenosine monophosphate, AMPK—AMP-activated protein kinase, mTORC1—Mechanistic target of rapamycin complex 1, mTORC2—Mechanistic target of rapamycin complex 2, MLST8 (MLST8 protein)—mammalian lethal with SEC13 protein 8, PRAS40—Proline-rich AKT substrate 40 kDa, Ras—Rat sarcoma protein, Raf—Rapidly accelerated fibrosarcoma protein, MEK—mitogen-activated protein kinase kinase, ERK—extracellular signal-regulated kinase, RSK—Ribosomal S6 kinase, PI3K—phosphoinositide 3-kinase, Akt—protein kinase B (Akt), TSC1/2—tuberous sclerosis complex 1/2, Rheb—Ras homolog enriched in brain, GDP—Guanosine diphosphate, MSIN1—MAPK (mitogen-activated protein kinase)-interacting protein 1, MLSTS (MLSTS protein)—mammalian lethal with SEC13 protein, Reg A/B (regulatory proteins A/B), GTP—Guanosine triphosphate, Reg C/D (regulatory proteins C/D), Treg—regulatory T cells, M1—M1 macrophages (classically activated macrophages), M2—M2 macrophages (alternatively activated macrophages), Th1—T helper 1 cells, Th17—T helper 17 cells. Figure 2 has been created in BioRender.com (accessed on 2 July 2024).

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
