# Peer review of "Diabetes and Osteoarthritis: Exploring the Interactions and Therapeutic Implications of Insulin, Metformin, and GLP-1-Based Interventions"

_biomedicines, 2024, doi:10.3390/biomedicines12081630_

Round 1
Reviewer 1 Report
Comments and Suggestions for Authors
The submitted manuscript summarized the relationships between osteoarthritis and diabetes to elucidate the underlying mechanism of osteoarthritis progression, and reviewed the therapeutic implications of insulin, metformin, and GLP-1-based interventions in osteoarthritis. Although this review is comprehensive and described in detail, there are still some major concerns.
1 The iThenticate analysis showed the match percent is so high to 44%. It is necessary to reduce the match percent.
2 Some subsections were not summarized well. For instance, the title of 2.1 is “Intersection of Type 1 Diabetes Mellitus and Osteoarthritis: Shared Mechanisms and Therapeutic Challenges”. However, in the text, authors only described their relationships and no mechanisms and therapeutic challenges were addressed.
3 In Section 2 and 3, more contents were related to diabetes, while less was related to osteoarthritis. So it is better to reorganize the text to address directly osteoarthritis from the relation with diabetes and the underlying mechanism as well as therapeutic implications of insulin, metformin, and GLP-1-based interventions.
4 Some sentences were confusing. For example, line 70-72 “However, there is limited information on how osteoarthritis and different types of diabetes, as well as the medications used to treat diabetes, influence the progression of osteoarthritis when both conditions are present.”
5 Some sentences were not provided the references. For example, there is only one reference (30) in the last paragraph in the page 3.
Thus I strongly suggest the authors carefully revise manuscript to improve the quality for easy understanding for readers.
Comments on the Quality of English LanguageSome sentences were confusing.
Author Response
We thank the Reviewer for the valuable feedback and suggestions. We have revised our article in accordance with the Reviewer’s recommendations. The changes have been highlighted in the text.
1. The iThenticate analysis showed the match percent is so high to 44%. It is necessary to reduce the match percent.
1. Response: This issue has been resolved.
2. Some subsections were not summarized well. For instance, the title of 2.1 is “Intersection of Type 1 Diabetes Mellitus and Osteoarthritis: Shared Mechanisms and Therapeutic Challenges”. However, in the text, authors only described their relationships and no mechanisms and therapeutic challenges were addressed.
2. Response: These subsections were now revised, and corrections were made to improve clarity and accuracy in summarizing the content.
3. In Sections 2 and 3, more contents were related to diabetes, while less was related to osteoarthritis. So it is better to reorganize the text to address directly osteoarthritis from the relation with diabetes and the underlying mechanism as well as therapeutic implications of insulin, metformin, and GLP-1-based interventions.
3. Response: Sections 2 and 3 have now been enriched with additional content focusing on osteoarthritis.
4. Some sentences were confusing. For example, line 70-72 “However, there is limited information on how osteoarthritis and different types of diabetes, as well as the medications used to treat diabetes, influence the progression of osteoarthritis when both conditions are present.”
4. Response: All of these observations were addressed. The ambiguities and inaccuracies have been corrected and clarified.
5. Some sentences were not provided the references. For example, there is only one reference (30) in the last paragraph in the page 3.
5. Response: We have added additional references to validate the assertions.
Reviewer 2 Report
Comments and Suggestions for Authors
Dear Authors,
The article titled "Diabetes and Osteoarthritis: Exploring the Interactions and Therapeutic Implications of Insulin, Metformin, and GLP-1-Based Interventions" provides an in-depth analysis of the complex interactions between diabetes mellitus (DM) and osteoarthritis (OA), focusing on the therapeutic implications of insulin, metformin, and GLP-1-based interventions. The authors have conducted a comprehensive and meticulous literature review grounded in the most recent research. However, several elements require improvement for the manuscript's full potential.
Minor Revisions:
- Keywords Adjustment: The keywords should differ from those in the title to enhance the article's discoverability and specificity in search databases.
- Formatting of In Vivo and In Vitro: Throughout the text, the terms "in vivo" and "in vitro" should be italicised to adhere to standard scientific formatting conventions.
- Explanation of Abbreviations Under Figures: All abbreviations used in the figures should be clearly explained to ensure readers can easily interpret the data presented.
Major Revisions:
- Distinction Between DM Type 1 and Type 2 in Introduction: The introduction must clearly differentiate between Type 1 Diabetes (T1DM) and Type 2 Diabetes (T2DM). T1DM is an autoimmune disease where obesity and insulin resistance are rare, whereas T2DM is closely linked with insulin resistance. It is crucial to separate the discussion of insulin resistance associated with T2DM from incidents of hyperglycemia in T1DM to avoid confusion.
- Clarification of Obesity Concepts: The terms "metabolically unhealthy obesity" and "metabolically healthy obesity" mentioned in line 124 require detailed explanation. What defines "metabolically healthy obesity"? How does it transition into "metabolically unhealthy obesity"? Clear criteria and descriptions should be provided.
- Precision in Insulin Secretion Statement: The statement in lines 128-129, "30% of insulin secretion is attributed to circulating free fatty acids in individuals both with and without diabetes," needs clarification. The current phrasing implies that insulin is required for the presence of free fatty acids, which may be misleading.
- Clarification of Estrogen and Cartilage Protection: The sentence in lines 180-181, "This sex difference may be attributed to decreased estrogen levels, which protect cartilage, after menopause," needs to be clarified. Does the reduction in estrogen levels protect cartilage? This contradicts previous cohort study descriptions and should be re-evaluated for coherence.
Author Response
We thank the Reviewer for insightful feedback and suggestions. We have revised our article in accordance with recommendations. The changes have been highlighted in the manuscript.
Minor Revisions:
1.Keywords Adjustment:The keywords should differ from those in the title to enhance the article's discoverability and specificity in search databases.
1. Response: The key terms were now revised to enhance the article's discoverability and specificity in search databases.
2. Formatting of In Vivo and In Vitro:Throughout the text, the terms "in vivo" and "in vitro" should be italicised to adhere to standard scientific formatting conventions.
2. Response: These comments were now considered and modified.
3. Explanation of Abbreviations Under Figures:All abbreviations used in the figures should be clearly explained to ensure readers can easily interpret the data presented.
3. Response: The abbreviations utilized in the figures and figure legends have now been clarified and explained.
Major Revisions:
1. Distinction Between DM Type 1 and Type 2 in Introduction:The introduction must clearly differentiate between Type 1 Diabetes (T1DM) and Type 2 Diabetes (T2DM). T1DM is an autoimmune disease where obesity and insulin resistance are rare, whereas T2DM is closely linked with insulin resistance. It is crucial to separate the discussion of insulin resistance associated with T2DM from incidents of hyperglycemia in T1DM to avoid confusion.
1. Response: We have now clarified the distinct differences between T1DM and T2DM.
2. Clarification of Obesity Concepts:The terms "metabolically unhealthy obesity" and "metabolically healthy obesity" mentioned in line 124 require detailed explanation. What defines "metabolically healthy obesity"? How does it transition into "metabolically unhealthy obesity"? Clear criteria and descriptions should be provided.
2. Response: The criteria for metabolically healthy and unhealthy obesity have now been clarified in the text.
3. Precision in Insulin Secretion Statement:The statement in lines 128-129, "30% of insulin secretion is attributed to circulating free fatty acids in individuals both with and without diabetes," needs clarification. The current phrasing implies that insulin is required for the presence of free fatty acids, which may be misleading.
3. Response: It has been now modified and clarified in the text.
4. Clarification of Estrogen and Cartilage Protection:The sentence in lines 180-181, "This sex difference may be attributed to decreased estrogen levels, which protect cartilage, after menopause," needs to be clarified. Does the reduction in estrogen levels protect cartilage? This contradicts previous cohort study descriptions and should be re-evaluated for coherence.
4. Response: It has been now modified and clarified.
Round 2
Reviewer 1 Report
Comments and Suggestions for Authors
After a major revision, the quality of this submitted manuscript has been improved greatly. I recomment it to be accepted.
Reviewer 2 Report
Comments and Suggestions for Authors
Dear Authors,
I accept the manuscript for publication in its present form.
Best regards,